# Isolation and Identification of Pennogenin Tetraglycoside from *Cestrum nocturnum* (Solanaceae) and Its Antifungal Activity against *Fusarium kuroshium*, Causal Agent of Fusarium Dieback

**DOI:** 10.3390/molecules27061860

**Published:** 2022-03-13

**Authors:** Erika Valencia-Mejía, Yeli Y. León-Wilchez, Juan L. Monribot-Villanueva, Mónica Ramírez-Vázquez, Israel Bonilla-Landa, José A. Guerrero-Analco

**Affiliations:** 1Laboratorio de Química de Productos Naturales, Red de Estudios Moleculares Avanzados, Instituto de Ecología A.C. (INECOL)—Clúster Científico y Tecnológico BioMimic^®^, Carretera Antigua a Coatepec N. 351, Xalapa 91073, Veracruz, Mexico; valenciamejiaerika@gmail.com (E.V.-M.); yesenia.leon.0319@gmail.com (Y.Y.L.-W.); juan.monribot@inecol.mx (J.L.M.-V.); 2Unidad de Microscopía Avanzada, Red de Estudios Moleculares Avanzados, Instituto de Ecología A.C. (INECOL)—Clúster Científico y Tecnológico BioMimic^®^, Carretera Antigua a Coatepec N. 351, Xalapa 91073, Veracruz, Mexico; m.ramirez@ciencias.unam.mx; 3Facultad de Ciencias, Universidad Nacional Autónoma de México (UNAM), Circuito Exterior, Cd. Universitaria, Copilco, Coyoacán, Ciudad de México 04510, Mexico; 4Laboratorio de Química Orgánica, Red de Estudios Moleculares Avanzados, Instituto de Ecología A.C. (INECOL)—Clúster Científico y Tecnológico BioMimic^®^, Carretera Antigua a Coatepec N. 351, Xalapa 91073, Veracruz, Mexico; israel.bonilla@inecol.mx

**Keywords:** *Fusarium kuroshium*, Fusarium dieback, antifungal activity, *Cestrum nocturnum*, steroidal saponins

## Abstract

Antifungal assay-guided fractionation of the methanolic crude extract of *Cestrum nocturnum* (Solanaceae), popular known as ‘lady of the night’, led the isolation and identification of the steroidal saponin named pennogenin tetraglycoside, which was identified for the first time in this plant species by spectroscopic means. The crude extract, fractions and pennogenin tetraglycoside exhibited mycelial growth inhibition of *Fusarium solani* and *F. kuroshium*. *F. solani* is a cosmopolitan fungal phytopathogen that affects several economically important crops. However, we highlight the antifungal activity displayed by pennogenin tetraglycoside against *F. kuroshium*, since it is the first plant natural product identified as active for this phytopathogen. This fungus along with its insect symbiont known as Kuroshio shot hole borer (*Euwallacea kuroshio*) are the causal agents of the plant disease Fusarium dieback that affects more than 300 plant species including avocado (*Persea americana*) among others of ecological relevance. Scanning electron microscopy showed morphological alterations of the fungal hyphae after exposure with the active fractions and 12 phenolic compounds were also identified by mass spectrometry dereplication as part of potential active molecules present in *C. nocturnum* leaves.

## 1. Introduction

Fusarium dieback (FD) is a rapidly spreading plant disease caused by an exotic ambrosia complex made up of a beetle and its symbiotic filamentous fungi. The female beetles dig into the host trees, creating galleries, and these galleries are inoculated with fungal spores, which are carried by the beetles in specialized structures called mycangia. The fungus colonizes the walls of the gallery, invading the vascular tissue of the tree, blocking water and nutrients transport causing wilting and, in most cases, the death of the tree, a few weeks after the infestation. This fungus-beetle complex uses as hosts more than 300 different plant species, among which are species of the Lauraceae family such as avocado (*Persea americana*), a species of economic importance for Mexico and worldwide [1,2,3]. 

FD is an emerging pest in the United States of America (USA) extending over much of its west coast territory [4,5,6]. In 2015, a new fungal species (*Fusarium kuroshium*) symbiotically associated with a borer beetle (*Euwallacea kuroshio*) was identified in Mexico (Tijuana, Baja California) but present since 2012, in California [7,8]. Currently, there is no effective method to control the spread of this complex pest, other than pruning (mechanical removal) of infected specimens to reduce beetle populations and the use of conventional synthetic agrochemicals, which causes ecological repercussions such as pollution of aquatic ecosystems, resistance of pest populations, among other undesirable effects [9,10,11,12]. 

Research efforts have considered biological control as an option to mitigate the negative impact of fungal diseases [13]. In this context, a good strategy is the study of natural products of botanical origin that have historically been considered effective for pest control as well as safer for human health and the environment, because of their short lifetime in the environment. One of the main sources of these natural products is plants present in tropical and subtropical forests as these are ecosystems with great biological diversity, and therefore phytochemical diversity [14].

In this regard, *Cestrum nocturnum* is a shrub of interest in the Solanaceae family, whose flowers exude a sweet fragrance at night, which is the main reason for its popular names such as lady of the night, night cestrum and night-blooming jessamine [15]. It is widely distributed in tropical and subtropical forests across the world, from Oceania (Australia) and Asia (China and India) to most of America, from USA to Brazil [16]. Several phytochemical studies have shown the presence of important antimicrobial compounds in different parts of the plant: saponins, alkaloids, flavonol glycosides, fatty acids, essential oils, phenolics and others [17].

Previous work carried out in the Natural Products Chemistry Laboratory at the BioMimic^®^ Cluster of the Institute of Ecology (INECOL by its acronym in Spanish) on the determination of antifungal potential of plant species present in the montane cloud forest of Veracruz, Mexico, allowed us to determine the preliminary in vitro antifungal potential against *F. kuroshium* of a crude methanolic extract of *C. nocturnum*, among other species (unpublished data). However, the metabolites responsible for this antifungal activity were unknown. Therefore, the main objective of the present work was to determine the antifungal activity against *F. kuroshium* of the methanolic crude extract of *C. nocturnum* and to identify the active compounds, contributing to the potential development of novel agents of botanical origin useful in the control of emerging pests such those involved in FD. For this, we carried out bioassay-guided fractionation studies using *F. solani* as a first microbiological model in common laboratory conditions and then some fractions were tested against *F. kuroshium* under biosecurity conditions. The use of *F. solani* strains as a first approach have been previously reported [18,19] due to its close phylogenetic relationship with symbiotic fungi of Euwallacea ambrosia beetles [20,21] and because *F. kuroshium* is kept under quarantine status by the Mexican phytosanitary authorities. 

## 2. Results

### 2.1. Extraction and Analysis of Phenolic Compounds by UPLC-MS-QqQ

Dried material (946.2 g) of aerial parts of *C. nocturnum* yielded a total of 64.7 g of methanolic crude extract (ECn; yield 6.84%). The UPLC-MS-QqQ analysis of ECn allowed us to identify and quantify 12 phenolic compounds, mainly phenolic acids (Table 1). L-phenylalanine and 4-hydroxybenzoic acid were the most abundant compounds identified, followed by *t*-cinnamic, salicylic and vanillic acids (Table 1). Kaempferide, vanillin, 4-hydroxyphenylacetic acid, *p*-anisic acid, secoisolariciresinol, *p*-coumaric acid and ferulic acid were also identified in smaller amounts (Table 1). 

### 2.2. Fractionation and Antifungal Activity

In the preliminary fractionation by liquid-liquid partitioning, five primary fractions corresponding to the solvents hexane (ECn-F1), dichloromethane (ECn-F2), ethyl acetate (ECn-F3), *n*-butanol (ECn-F4) and water (ECn-F5) were obtained and used to test their antifungal activity against *F. solani* at a concentration of 2 mg mL^−1^. All primary fractions inhibited the mycelial growth of this fungus three days after inoculation and the effect was more noticeable after seven days (Figure 1a). The hexane (ECn-F1) and butanolic (ECn-F4) fractions exhibited the highest inhibition three days after inoculation with percentage values of 41.9 and 49.7, respectively. The ECn-F4 fraction exhibited the highest inhibition after seven days post-inoculation (Figure 1a). Based on these results, ECn-F4 fraction was further subfractionated obtaining 25 secondary fractions of which 20 were evaluated in the antifungal activity assay against *F. solani* (Figure 1b). The secondary fractions ECn-F4-16, ECn-F4-17 and ECn-F4-19 exhibited 100% inhibition, the same as the positive control (thiabendazole) at the same concentration. Thiabendazole is a commercial chemical compound with a broad-spectrum of antifungal activity.

Considering the obtained amount for each subfraction from the second fractionation as well as the results from the antifungal activity assay against *F. solani* (Figure 1), seven secondary fractions were selected for further evaluation against *F. kuroshium* in the Mexican National Phytosanitary Reference Center (CNRF by its acronym in Spanish). Five secondary fractions exhibited high inhibition of *F. kuroshium* mycelial growth (Figure 2a), corroborating the antifungal potential of the secondary fractions ECn-F4-3, ECn-F4-16, ECn-F4-17 and ECn-F4-19, which exhibited the highest growth inhibition of *F. solani* (Figure 1b). On the other hand, the fractions ECn-F4-18, ECn-F4-20 and ECn-F4-25, were not remarkably active against *F. solani* (Figure 1b) but were also considered as promising against *F. kuroshium* (Figure 2a). The secondary fractions that exhibited the highest inhibition of the mycelial growth of *F. kuroshium* were evaluated at lower concentrations (1 mg mL^−1^, 0.5 mg mL^−1^ and 0.2 mg mL^−1^). ECn-F4-17 and ECn-F4-25 inhibited 100% the growth of *F. kuroshium* at the lowest concentration evaluated (0.2 mg mL^−1^) (Figure 2b), highlighting the fact that one of them consisted of an amorphous precipitate (ECn-F4-25), which was obtained from the eluates 181–191 and the resulting liquor was identified as ECn-F4-17.

### 2.3. Microscopic Analysis

In order to obtain a view of the morphological effects of some selected ECn-F4 secondary antifungal active fractions on the *F. kuroshium* mycelium, microscopic observations were made using a scanning electron microscopy platform (SEM). The secondary fractions ECn-F4-16 (at 0.5 mg mL^−1^), ECn-F4-18 (at 0.2 mg mL^−1^) and ECn-F4-20 (at 0.5 mg mL^−1^) exhibited moderate mycelial growth inhibition (Figure 2b) and provoked deformations in the mycelium of the fungus such as wrinkles in the hyphae, agglomerations, thinning, conglobations and expanded widths along the hyphae after six days of exposure (Figure 3).

### 2.4. Spectroscopic and Spectrometric Analyses

The amorphous precipitate present in the active ECn-F4-25 secondary fraction was analyzed by high resolution mass spectrometry (HRMS) and nuclear magnetic resonance (NMR) approaches, which allowed us to identify it as (25*R*)-5-enspirostan-3-β, 17α-diol-3-o-α-l-rhamnopyranosyl-(1-4)-α-*l*-rhamnopyranosyl-(1-4)-[α-*l*-rhamnopyranosyl-(1-)]-β-d-glucopyranoside, also known as pennogenin tetraglycoside. The mass spectrum (Figure 4) revealed a mass-charge ratio (*m*/*z*) of 1053.5240 Da [M+Na]^+^, which corresponds to the empirical formula assignable to C_51_H_82_O_21_Na^+^ (mass error: −0.6 ppm). In addition, four important mass losses were observed, three of them of 146 Da corresponding to three deoxyhexose (rhamnose) sugar moieties, and another one assignable to a hexopyranose (glucose) with mass loss of 162 Da. The *m*/*z* ratio found for the aglycone portion was 413.3055 Da.

To the best of our knowledge, all the previous studies about either identification or chemical shifts assignments of all protons and carbons present in the pennogenin tetraglycoside by NMR spectroscopy refer to the Nohara, et al. report [22], where the molecule was not fully described. In addition, most of the assignments for saccharide portions in saponins are performed through chemical derivatization, hydrolysis, reductions, oxidations and so on. Here, we performed comprehensive NMR studies to establish all the assignments for protons and carbons and the use of more advanced NMR pulse sequences allowed us to characterize the whole saponin without derivatization. The differences in the core structure of saponins lies in the number of carbons that are part of the aglycone, 27 carbon atoms for the steroid type which form a tetracyclic molecule and 30 carbon atoms for the triterpenoid type which form either tetra or a pentacyclic molecule. The 413.3055 *m*/*z* value found for the aglycone portion (Figure 4) suggest a steroidal-type saponin (diosgenin-like) with a complex hydrocarbon structure that it can be subdivided into the spirostane-type steroidal portion or sapogenin bound to one or more sugar moieties. However, the NMR spectroscopic data obtained (Table 2) exhibited an additional quaternary carbon at 90.71 ppm (C-17), a shift that can be justified if this carbon is the base of a hydroxyl group. Additionally, the position of the quaternary carbon was corroborated through the correlations observed in the long distance heteronuclear multiple bond correlation (HMBC) experiment (Appendix A), since correlations were observed with protons at H-16, H-18 and H-21, suggesting a spirostanol-type structure. These molecules are characterized as containing four methyl groups bonded to the main skeleton. The signals found at 19.93 (C-19), 17.72 (C-27), 17.62 (C-18) and 9.95 (C-21) ppm confirm the proposed hypothesis of a steroidal-type saponin (diosgenin-like).

The signal present at 3.89 ppm in the ^1^H spectrum (Appendix A) was assigned to the proton of carbon C-3 by the heteronuclear single quantum correlation (HSQC) experiment (Appendix A), allowing the assignment of the protons of the carbons C-4, C-5 and C-6 using the correlations found in the correlation spectroscopy (COSY) and total correlation spectroscopy (TOCSY) two-dimensional experiments (Appendix A). The signal at 5.36 ppm as a multiplet was assigned to the H-6 position allylic proton, allowing the assignment of the quaternary carbon at 141.47 ppm as an endocyclic double bond (C-5). The signals of carbons 7 and 8 were assigned directly through the correlations found in the HMBC experiment (Appendix A). Likewise, the assignments of quaternary carbons C-9 and C-10 were obtained through their correlations with the methyl group at C-19. Because the hydrogen signal of carbon 16 was in the glycosidic region, a selective irradiation TOCSY one dimensional experiment to the β proton of carbon 15 was carried out, which allowed to assign the protons and carbons of positions C-14 and C-16.

The tetrahydropyran ring was assigned using the signal corresponding to the methyl group of carbon C-27, which through the COSY experiment (Appendix A) allowed to assign the signals of the C-26 protons. The remaining protons were assigned using their COSY correlations (Appendix A). However, due to the high overlap of the signals from the protons at positions C-23 and C-24, a heteronuclear 2-bond correlation (H2BC) experiment (Appendix A) was used to differentiate between the couplings at two and three distance bonds, making the assignment of the mentioned signals unequivocal. All the data analyzed led us to the conclusion of assigning the main nucleus as pennogenin-type compound.

The next step was the assignment of all signals of the glycosidic part. The irradiation of all glycosidic protons let us to assign all monosaccharides signals (Table 3) and confirmed the mass losses obtained from the HRMS (Figure 4). Finally, in order to determine the sequence and position for each monosaccharide moiety the correlations found in the HMBC experiment were used, observing that the anomeric carbon of glucose (C-1) (Table 3) in 100.94 ppm correlated with the proton located in H-3 of the aglycone. In the same way, correlations were found between the carbon C2 and C-4 carbon of glucose with the anomeric protons of two rhamnose units at 5.69 ppm (H-1, rhamnose I) and 6.23 ppm (H-1, rhamnose III). Therefore, a correlation was also observed between the proton at 6.14 ppm corresponding to another unit of rhamnose II with the C-4 at 80.87 ppm of rhamnose I. Finally, the stereochemistry was confirmed by the one-dimensional rotating frame overhause effect spectroscopy (ROESY 1D) experiment. Figure 5 summarizes all correlations determined by ROESY (red double-headed arrows), HMBC (black single-headed arrows) and COSY (black lines).

## 3. Discussion

*Cestrum nocturnum* is a well-known plant for its fragrant flowers and volatiles compounds present in the essential oils [16,23,24]. However, few phytochemical studies have been performed targeting the presence of bioactive molecules in more polar extracts [17,25,26,27]. In the present work, 12 phenolic compounds were identified and quantified in a methanolic extract of aerial parts of *C. nocturnum* (Table 1). From these, only vanillic and salicylic acids were previously identified in a methanolic extract of *C. nocturnum* flowers [26]. Kaempferol and some derivatives, including kaempferol 8-*O*-methyl ether, were previously reported in chloroform and ethyl acetate subfractions of a methanolic extract from aerial parts of *C. nocturnum* [27]. In our study, we identified kaempferide which is kaempferol 4-*O*-methyl ether and had not been previously described in this species (Table 1). Some of the identified phenolics have been reported with antifungal activity. For example, 4-hydroxybenzoic acid exhibited antifungal activity against *Epidermophyton floccosum* (MIC = 0.54 mg mL^−1^) and *Trichophyton rubrum* (MIC = 0.54 mg mL^−1^) [28], *Saccharomyces cerevisiae* (IC_25_ = 1380 µg mL^−1^) and *Candida albicans* (IC_25_ = 1380 µg mL^−1^) [29]. Cinnamic acid and some synthetic derivatives such as that with substitutions on the cinnamic acid phenyl ring with a trifluoromethyl group, exhibited antifungal activity at 0.5 mmol L^−1^ against *Cochliobolus lunatus*, *Aspergillus niger* and *Pleurotus ostreatus* by inhibiting benzoate 4-hydroxylase (CYP53A15), a key enzyme involved in benzoate detoxification [30]. Also, *t*-cinnamic acid inhibits the growth of *Fusicladium effusum* (64 µg mL^−1^), *Colletotrichum acutatum* (10 µg), *C. fragariae* (10 µg) and *C. gloesporioides* (10 µg) [31], while salicylic acid inhibited the mycelial growth of *Eutypa lata* (at 1 mM) [32] and *Penicillium expansum* (at 2.5 mM) [33]. Hence, the phenolics found in the ECn of *C. nocturnum* could be co-responsible in part of the antifungal effect observed for this plant species. 

Nonetheless, with the goal of identifying additional bioactive molecules present in *C. nocturnum* active extract, the antifungal assay-guided fractionation was also carried out. The ECn and fractions (primary and secondary) exhibited antifungal activity, inhibiting the mycelial growth of both tested fungi (*F. solani* and *F. kuroshium*) in a range between 6–100% (Figure 1 and Figure 2). These results agreed with previous studies of the antifungal activity of *C. nocturnum* performed in similar ways, such as the one carried out by Al-Reza et al. [24], who reported the antifungal effect after 5–6 days of the essential oil (at 1000 ppm) and organic extracts (hexane, chloroform, ethyl acetate and methanol extracts at 1500 μg/disc) of *C. nocturnum* against the phytopathogenic fungi *Botrytis cinerea, Colletotrichum capsici, Fusarium oxysporum, F. solani, Phytophthora capsici, Rhizoctonia solani* and *Sclerotinia sclerotiorum*, highlighting the percentage of inhibition of 78.8% against the *F. solani* strain 41092 from the Korean Agricultural Culture Collection (KACC). The acetone, hexane and chloroform fractions exhibited a greater antifungal effect against the *F. oxysporum* strain 41083 from KACC, a phylogenetically related fungus with *F. solani*. Similarly, Bautista-Baños et al. [34], evaluated the fungicidal potential of *C. nocturnum* organic extracts on two Fusarium isolates from *Carica papaya* and *Spondias purpurea*, where the methanolic extract inhibited the mycelial growth and sporulation of both isolates at 12 mg mL^−1^ after 10 days. Finally, Rashed et al. [27] reported that the ethyl acetate fraction obtained from a methanolic extract from aerial parts of *C. nocturnum* exhibited the highest inhibitory and fungicidal activities at 0.075–0.3 mg mL^−1^ and 0.15–0.6 mg-mL^−1^, respectively, against *Aspergillus fumigatus* (ATCC 1022), *Aspergillus versicolor* (ATCC 11730), *Aspergillus ochraceus* (ATCC 12066), *A. niger* (ATCC 6275), *Trichoderma viride* (IAM 5061), *Penicillium funiculosum* (ATCC 36839), *Penicillium ochrochloron* (ATCC 9112) and *Penicillium verrusocum var. cyclopium* after 3 days of exposure. Although the antifungal activity of *C. nocturnum* compounds against our first microbiological model (*F. solani*) has already been reported in the literature, there were no previous reports of inhibition of *F. kuroshium* by plant natural products until now. 

In this work, pennogenin tetraglycoside (a steroidal saponin) was identified as the main compound responsible for the antifungal activity displayed by the crude extract and fractions of *C. nocturnum*. This molecule was previously identified by Kang et al. [35] from the rhizomes of *Paris polyphylla* along with other four saponin-type compounds. In the literature there are previous reports about the antifungal activity exhibited by spirostan type saponins. For example, pennogenin steroidal saponins isolated from the ethanolic extract of *P. polyphylla* var. *yunnanensis*, exhibited moderate antifungal activity against *S. cerevisiae* and *C. albicans* [36] and the spirostanol saponin (25R)-1β,2α-dihydroxy-5α-spirostan-3-β-yl-o-α-l-rhamnopyranosyl-(1→2)-β-d-galactopyranoside present in *Cestrum schlechtendahlii* (at 0.5 mg/disk) inhibited the growth of *S. cerevisiae*, *C. albicans*, *Cryptococcus neoformans*, and *F. graminearum* [37]. In addition, other biological activities have been also reported for spirostan derivatives. Fu et al. [38] identified pennogenin glycosides as the active principles of *P. polyphylla* var. *yunnanensis* in promoting hemostasis *in vivo*, representing a new type of platelet aggregation inducer; Namba et al. [39] described diosgenin-3-*o*-α-l-rhamnopyranosyl-(1→2)-(α-l-arabinofuranosyl-(1→4))-d-glucopranoside with chronotropic effects on spontaneous beating of cells and Matsuda et al. [40] reported four spirostanol-type steroidal saponins present in *P. polyphylla* species with protective effects on gastric mucosal lesions.

The precise mechanism of action that explains the antifungal activity exhibited by *C. nocturnum* extract and fractions is not fully understood. However, Sharma and Tripathi [41] suggested that the components of some botanical extracts, specifically those with saponin compounds which present amphipathic chemical features would act on the hyphae of the mycelium causing the release of the components of the cells cytoplasm and/or the loss of rigidity and integrity of the cells wall, leading to the collapse and death due their well-known surfactant properties. This hypothesis agrees with our results, where severe deformations of the hyphae were determined by SEM (Figure 3). Further studies are needed to address the mechanism of action of the active compounds here identified.

## 4. Materials and Methods

### 4.1. Plant Material

The aerial parts of the *C. nocturnum* plant were collected in the “Santuario del Bosque de Niebla”, a natural protected area of the ecological reserve “Francisco Javier Clavijero” of the INECOL at Xalapa, Veracruz, Mexico. The taxonomic identification was confirmed by Dr. Sergio Avendaño Reyes, curator of the INECOL XAL Herbarium, where a voucher was deposited (149040 and 149041).

### 4.2. Fungal Strains

The strain of *F. solani* used in this study was kindly provided by Dr. Mauricio Luna-Rodríguez (Faculty of Agricultural Sciences of the Universidad Veracruzana) and it was isolated from chilli (*Capsicum annum* L.) [18]. The strain number is LAT-059 and it is deposited in the collection of the High Technology Laboratory of Xalapa of the Universidad Veracruzana (LATEX by its acronym in Spanish). The strain of *F. kuroshium* HFEW-16-IV-019 was kindly provided by the National Service for Agrifood Health, Safety and Quality (SENASICA by its acronym in Spanish), through the Mexican CNRF, where the experiments were carried out at biosecurity facilities since this phytopathogen is categorized as a quarantined organism by the Mexican phytosanitary authorities. The fungus was isolated from the ambrosia beetle *Euwallacea* sp. nr. *fornicatus* now renamed as *E. kuroshio* (Kuroshio shot hole borer, KSHB) [7]. Strains were periodically seeded in potato dextrose agar medium and incubated in the dark at 27 ± 2 °C for seven days prior antifungal assays.

### 4.3. Preparation of the Methanolic Crude Extract

Leaves and stems of *C. nocturnum* were washed with distilled water, cut into small pieces and frozen for 24 h at −80 °C (Thermo Scientific, 900 series, Waltham, MA, USA). Subsequently, the material was lyophilized (Labconco, Freezone 1, Kansas City, MO, USA) and pulverized in a blade mill (Pulverissette 15, Fritsch GmbH, Idar-Oberstein, Germany). Then the material (946.2 g) was subjected to an extraction process by maceration with methanol (MeOH) in a 1:10 weight/volume ratio at room temperature under constant stirring for 24 h. This procedure was repeated three times with the resulting residual material. Finally, supernatants of the extractions were combined, filtered and the solvent was eliminated by rotary evaporation under reduced pressure (Buchi, RII, Flawil, Switzerland) to yield 64.7 g of ECn.

### 4.4. Analysis of Phenolic Compounds by UPLC-MS-QqQ

The identification and quantification of individual phenolic compounds in ECn was performed by ultra-high-performance liquid chromatography (Agilent, 1290, Santa Clara, CA, USA) coupled to a triple quadrupole mass spectrometer (Agilent, 6460, Santa Clara, CA, USA; UPLC-MS-QqQ) with a dynamic multiple reaction monitoring method as is shown in Appendix A and as we previously described [42,43]. Chromatographic analysis was carried out using a reversed-phase column (Zorbax SB-C18; 1.8 μm, 2.1 × 50 mm; Agilent, Santa Clara, CA, USA) with the column oven temperature at 40 °C. The mobile phase (flow rate of 0.1 mL min^−1^) consisted of water (A) and acetonitrile (B), both containing formic acid (0.1%). The gradient conditions were: 0 min 1% B, 0.1–40 min linear gradient 1–40% B, 40.1–42 min linear gradient 40–90% B, 42.1–44 min isocratic 90% B, 44.1–46 min linear gradient 90–1% B, 46.1–47 min 1% B isocratic. The spectrometric analyses were conducted by electrospray ionization in positive and negative modes. For this, the capillary and injector voltages were 3500 and 500 V, respectively. The desolvation temperature was 300 °C, the sheath gas (N_2_) flow was 5 L min^−1^ and temperature was 250 °C, and the nebulizer pressure was 45 Psi. The fragmentor voltage was 100 V, and the collision energy was optimized individually for each compound. The identity of each compound was corroborated by co-elution with authentic commercial standards under the same analytical conditions. A calibration curve in a concentration range of 0.5–17 μM was prepared for quantitation of each compound and each point was injected in triplicate. Quadratic regressions were applied obtaining values of R^2^ ≥ 0.97 for the quantitation range. The data were processed using the MassHunter Workstation software, version B.06.00 (Agilent, Santa Clara, CA, USA) and the results were expressed as μg g^−1^ of dried plant material. 

### 4.5. Fractionation

#### 4.5.1. Preliminary Fractionation of the ECn

The primary fractionation was performed by liquid-liquid extraction three, using 60 g of the ECn resuspended in 1 L of water and extracted in an ascending polarity ratio (1:1, three times with each solvent) with hexane (ECn-F1), followed by dichloromethane (ECn-F2), ethyl acetate (ECn-F3), *n*-butanol (ECn-F4) and water (ECn-F5). The organic extracts (ECn-F1, Ecn-F2, ECn-F3 and ECn-F4) were filtered over sodium sulfate anhydrous (Na_2_SO_4_) and concentrated by rotary evaporation under reduced pressure (Buchi, RII). The remaining aqueous phase (ECn-F5) was lyophilized until dryness.

#### 4.5.2. Subfractionation of the Active Fraction ECn-F4

The secondary fractionation of ECn-F4 was carried out by affinity chromatography on an open glass column packed with silica gel (SiO_2_) as stationary phase. Previously, for the selection of the initial mobile phase, an analysis was carried out by thin-layer chromatography (TLC) visualized with an UV light lamp at λ = 254 and 325 nm, determining the initial mobile phase as a mixture of hexane and ethyl acetate 50:50 ratio. Subsequently, 6.5 g of the ECn-F4 fraction were adsorbed on 40 g of silica gel. The elution of the column was performed by adding 200 mL of a hexane-ethyl acetate 50:50 mixture, increasing the polarity in an ascending polarity gradient of 5%, passing through 100% of ethyl acetate until reaching ethyl acetate-methanol 50:50 and a final elution step with 100% methanol, yielding a total of 270 eluates that were pooled on the basis of their TLC profiles to obtain 24 secondary fractions (ECn-F4-1 to ECn-F4-24). Additionally, during the solvent evaporation process, an amorphous beige powder precipitated spontaneously from eluates 181 to 210 which was separated by decantation and then included as ECn-F4-25 in the antifungal activity evaluations. 

### 4.6. In Vitro Evaluation of the Antifungal Activity 

A spore solution was made by immersing the mycelium of *F. solani* or *F. kuroshium* in 10 mL of sterile distilled water. The solution was homogenized by shaking vigorously and the spores were quantified using a hemocytometer (Bright-LineTM Hemacytometer, Sigma^®^, St. Louis, MO, USA) adjusting them to a concentration of 1 × 10^6^ spores mL^−1^. The antifungal activity of the extract and the fractions of *C. nocturnum* was evaluated as follows: the treatments were resuspended in a vehicle (H_2_O-MeOH in a 15:85 ratio) and mixed with PDA medium. The 12-well plates were prepared by placing 1 mL of PDA with the treatment volume adjusted to the concentrations (0.2, 0.5, 1 and 2 mg mL^−1^) to be evaluated as well as the controls (positive control: thiabendazole (2 mg mL^−1^); negative control (vehicle). Once the medium solidified, 3 μL of the spore solution was placed in the center of each well, then the plates were incubated at 27 °C for 3, 5 and 7 days. Mycelial growth diameter was measured once every 24 h, using Mycometre software 2.01 (Georges Fannechere). Experiments were performed in triplicate.

Inhibition percentage of mycelial growth was calculated as follows:$$\mathrm{Mycelial}\;\mathrm{growth}\;\mathrm{inhibition}\;(\%)=\frac{\mathrm{DC}\;-\;\mathrm{DT}}{\mathrm{DC}}\;\times \;100$$
where DC and DT are the average diameters in mm of fungal mycelia of control and treatments, respectively.

### 4.7. Microscopy Analysis

Mycelial samples of the *F. kuroshium* exposed to the secondary fractions of *C. nocturnum* (at 0.2 and 0.5 mg mL^−1^) were observed after six days of treatment in a SEM platform to evaluate the damage. The hyphae were fixed in Karnovsky’s solution, washing 2 to 3 times with phosphate-buffered saline buffer for five minutes to remove the excess. Washed samples were dehydrated in a graded ethanol series (30–100%) for 50 min for each concentration, dried in a critical point dryer (Quorum, K850, London, UK) with CO_2_ and attached to aluminum stubs using a carbon adhesive prior to coating with gold in a sputter coater (Quorum, Q150, London, UK). Preparations were observed and photographed in a microscope FEI Quanta 250-FEG (Hillsborough, OR, USA), at 5000× and 15,000×. 

### 4.8. Spectroscopic and Spectrometric Analyses of the Active Fraction ECn-F4-25

#### 4.8.1. High-Resolution Mass Spectrometric Analysis

The high-resolution mass spectra of the active fraction ECn-F4-25 (further identified as the pure compound pennogenin tetraglycoside) was performed by ultra-high-performance liquid chromatography (Waters, Class I, Milford, MA, USA) coupled to a quadrupole-time of flight mass spectrometer (Waters, Synapt G2-Si, Milford, MA, USA; UPLC-MS-QTOF). Chromatographic analysis was carried out using a reversed-phase column (Acquity BEH; 1.7 μm, 2.1 × 50 mm; Waters, Milford, MA, USA) with the column oven temperature at 40 °C. The mobile phase (flow rate of 0.3 mL min^−1^) consisted of water (A) and acetonitrile (B), both containing formic acid (0.1%). The gradient conditions were: 0–13 min linear gradient 1–80% B, 13–14 min isocratic 80% B, 14–15 min linear gradient 80–1% B, 15–20 min 1% B isocratic. The spectrometric analysis was conducted by electrospray ionization in positive mode. For this, the capillary, sampling cone and source offset voltages were 3000, 40 and 80 V, respectively. The source and desolvation temperature were 120 and 450 °C, respectively. The desolvation gas (N_2_) flow was 800 L h^−1^ and the nebulizer pressure was 6.5 Bar. Leucine-enkephalin was used as the lock mass (556.2771, [M+H]^+^. It was used a MS^e^ method with a mass range of 50-1200 Da. The energy of collision were 6 V for function 1 and a ramp of 10–30 V for function 2. The scan time was 0.5 s. The data were processed with MassLynx 4.1 (Waters, Milford, MA, USA).

#### 4.8.2. Spectroscopic Analysis of Isolated Pure Compound

The ECn-F4-25 (10 mg) fraction identified as the pure compound pennogenin tetraglycosed was dissolved in 0.5 mL of pyridine-*d*^5^ and transferred to a 5 mm borosilicate tube. All NMR experiments were recorded on a Bruker Avance III HD equipment (Billerica, MA, USA) at 500 MHz for ^1^H and 125 MHz for ^13^C equipped with a broadband observe (BBO) probe at 50 °C. The hard 90° H pulse was calibrated before the spectra acquisition with a value of 10.63 µs. The ^1^H experiment was recorded using water suppression (O1P = 4.99 ppm) under the ZGCPPR pulse sequence, both ^1^H and ^13^C were referenced using the residual signal of pyridine-*d*^5^ at 7.22 ppm (^1^H) and 123.5 ppm (^13^C). The 2D-HMBC experiment was optimized at a constant of 10 Hz. The 2D-TOCSY experiment was recorded using a mixing time of 80 ms. 1D-TOCSY experiment was recorded using the pulse sequence SELDIGPZS and a mixing time of 240 ms. The 1D-ROESY experiment was recorded using the pulse sequence SELROGP and optimized for a mixing time of 200 ms. All the spectra were processed using TopSpin 4.1.3 from Bruker Biospin^®^.

### 4.9. Statistical Analyses

Antifungal activity results were expressed as means ± standard deviation (*n* = 3). The data were analyzed using the non-parametric Kruskall-Wallis test followed by Mann-Whitney Post-hoc tests with adjustment by the Bonferroni method to show the significant differences in antifungal activity. Statistical analyses were done with R-studio software [44].

## 5. Conclusions

The results of this work demonstrated the relevant antifungal potential of the crude extract, fractions and one pure compound from *C. nocturnum* against two species of phytopathogenic fungi of the genus Fusarium (*F. solani* and *F. kuroshium*). The study highlights the biological importance of including plant species with high presence in the everwet montane cloud forest of Veracruz, Mexico for searching new antifungal natural products. In addition, 12 phenolic compounds were also identified and quantified in the active crude extract of *C. nocturnum* leaves, whose presence may be correlated with part of the biological effect observed at the extract level. Finally, the presence of a steroidal saponin-type compound was confirmed and identified as the main antifungal compound against *F. kuroshium*. The compound identified as (25R)-5-enspirostan-3-β,17α-diol-3-*o*-α-l-rhamnopyranosyl-(1-4)-α-l-rhamnopyranosyl-(1-4) -[α- l-rhamnopyranosyl-(1-2)]-β-d-glucopyranoside or by its common name pennogenin tetraglycoside, is reported for first time in the species *C. nocturnum* and is the first phytochemical reported able of inhibiting the growth of *F. kuroshium*, the causal agent of Fusarium dieback. Our findings highlight the importance of considering botanical natural products in the search for potential biocontrol agents in order to detect new bioactive compounds against persistent and emergent phytopathogenic fungi.

## Figures and Tables

**Figure 1 molecules-27-01860-f001:**
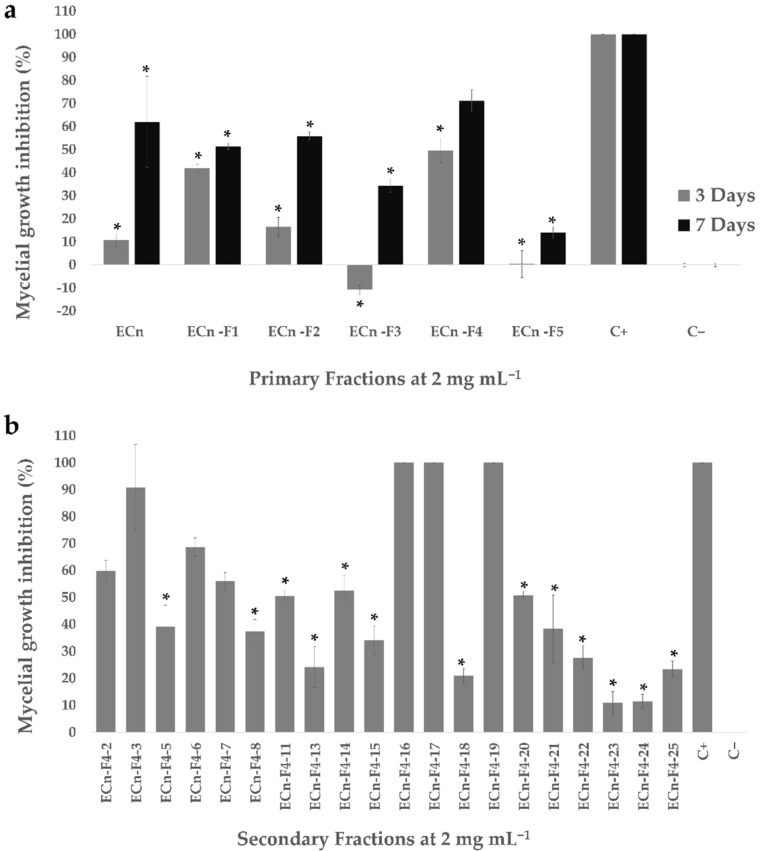
Mycelial growth inhibition displayed by primary and secondary fractions of *C. nocturnum* against *F. solani*. (**a**) Percentage of inhibition of *F. solani* growth exposed to primary fractions (2 mg mL^−1^) evaluated three and seven days post-inoculation. Data are the average ± standard error (*n* = 3). Asterisk (*) indicates significant difference compared to positive control (thiabendazole 2 mg mL^−1^; C+), *p* < 0.05, Kruskall-Wallis, Mann-Whitney post-hoc test with Bonferroni adjustment. The negative control (C−) was the vehicle (H_2_O-MeOH in a 15:85 ratio) where samples were dissolved. Appendix A is shown in Appendix A. (**b**) Percentage of inhibition of *F. solani* growth exposed to secondary fractions from ECn-F4 (2 mg mL^−1^). Data are the average ± standard error (*n* = 3). Asterisk (*) indicates significant difference compared to positive control (thiabendazole 2 mg mL^−1^; C+), *p* < 0.05, Kruskall-Wallis, Mann-Whitney post-hoc test with Bonferroni adjustment. The negative control (C−) was the vehicle (H_2_O-MeOH in a 15:85 ratio) where samples were dissolved. Appendix A is shown in Appendix A.

**Figure 2 molecules-27-01860-f002:**
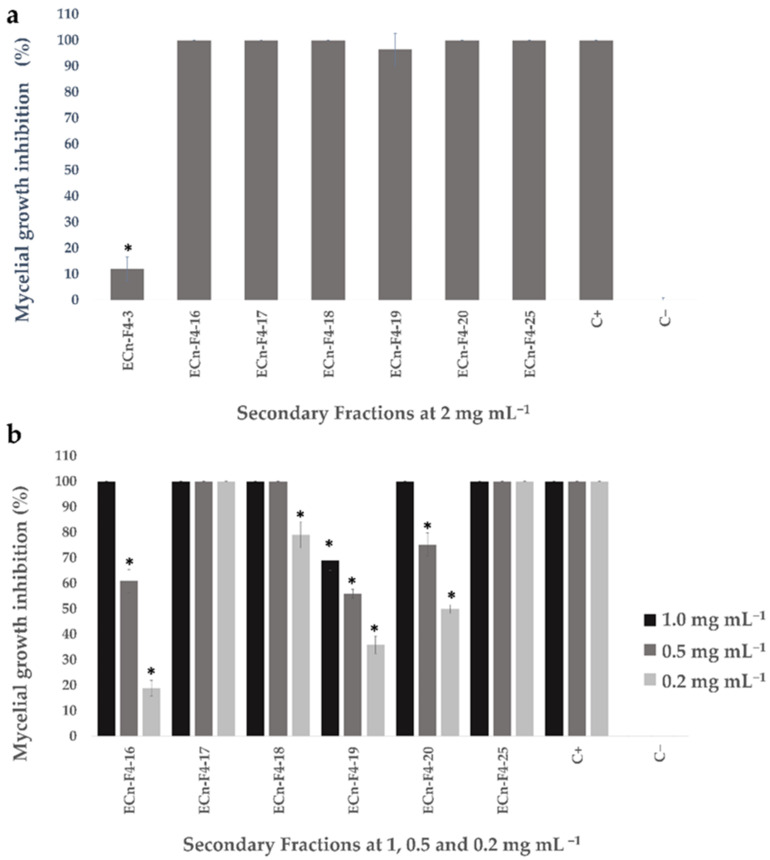
Mycelial growth inhibition displayed by secondary fractions from ECn-F4 against *F. kuroshium*. (**a**) Percentage of inhibition of *F. kuroshium* exposed to ECn-F4 secondary fractions at 2 mg mL^−1^. Data are the average ± standard error (*n* = 3). Asterisk (*) indicates significant difference compared to positive control (thiabendazole 2 mg mL^−1^; C+), *p* < 0.05, Kruskall-Wallis, Mann-Whitney post-hoc test with Bonferroni adjustment. The negative control (C−) was the extract vehicle (H_2_O-MeOH in a 15:85 ratio). Appendix A is shown in Appendix A. (**b**) Percentage of inhibition of *F. kuroshium* exposed to secondary fractions at 1, 0.5 and 0.2 mg mL^−1^. Data are the average ± standard error (*n* = 3). Asterisk (*) indicates significant difference compared to positive control (thiabendazole 2 mg mL^−1^; C+), *p* < 0.05, Kruskall-Wallis, Mann-Whitney post-hoc test with Bonferroni adjustment. The negative control (C−) was the vehicle (H_2_O-MeOH in a 15:85 ratio) where samples were dissolved. Appendix A is shown in Appendix A.

**Figure 3 molecules-27-01860-f003:**
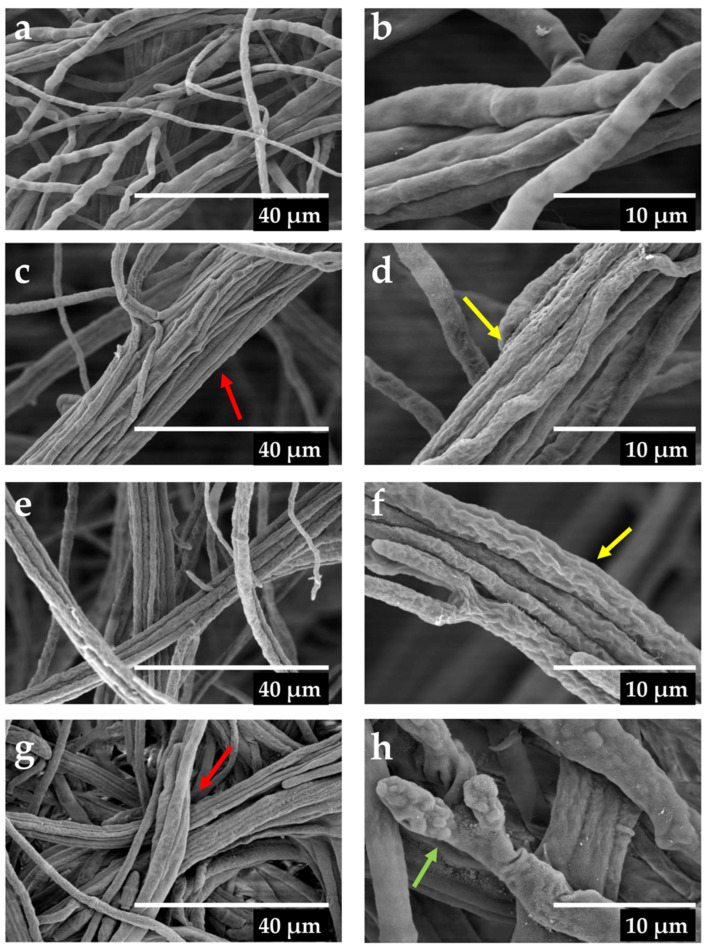
Morphological alterations in hyphae of *F. kuroshium* observed by scanning electron microscopy (SEM) caused by the secondary fractions of *C. nocturnum* after six days of exposure. (**a**) control (extract vehicle: H_2_O-MeOH in a 15:85 ratio) at 5000×. (**b**) control (extract vehicle: H_2_O-MeOH in a 15:85 ratio) at 15,000×. (**c**) ECn-F4-16 (0.5 mg mL^−1^) at 5000×. (**d**) ECn-F4-16 (0.5 mg mL^−1^) at 15,000×. (**e**) ECn-F4-18 (0.2 mg mL^−1^) at 5000×. (**f**) ECn-F4-18 (0.2 mg mL^−1^) at 15,000×. (**g**) ECn-F4-20 (0.5 mg mL^−1^) at 5000×. (**h**) ECn-F4-20 (0.5 mg mL^−1^) at 15,000×. Yellow, red and green arrows show wrinkles, agglomerations and conglobations, respectively.

**Figure 4 molecules-27-01860-f004:**
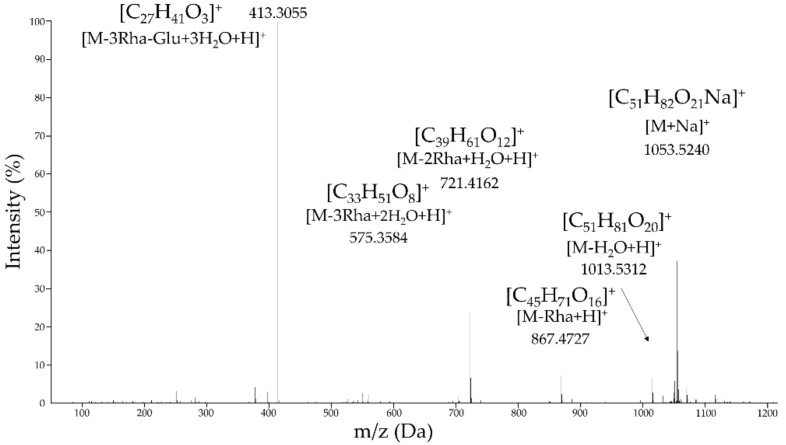
High resolution mass spectrum (ESI^+^) of the amorphous precipitate which corresponds to ECn-F4-25. The molecular ion and fragmentation pattern of the pure compound is showed. The maximum permissible mass error for elemental analysis was below ±5 ppm.

**Figure 5 molecules-27-01860-f005:**
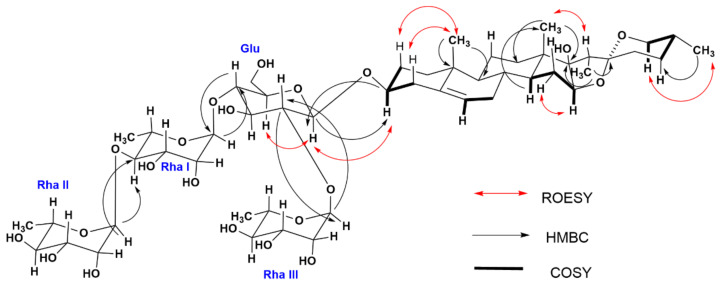
Summary of spectroscopic (NMR) results of the amorphous precipitate which corresponds to ECn-F4-25. Key correlations determined by COSY (black lines), HMBC (black single-headed arrows) and ROESY (red double-headed arrows) experiments for pennogenin tetraglycoside. Appendix A are explained in the text and provided in the Appendix A.

**Table 1 molecules-27-01860-t001:** Phenolic compounds identified and quantified in methanolic extract from *C. nocturnum*.

Compound Name	Content (μg g^−1^)
L-Phenylalanine	12.47 ± 0.07
4-Hydroxybenzoic acid	10.48 ± 0.10
*t*-Cinnamic acid	2.99 ± 0.06
Salicylic acid	2.36 ± 0.02
Vanillic acid	1.19 ± 0.02
Kaempferide	0.58 ± 0.01
Vanillin	0.56 ± 0.00
4-Hydroxyphenylacetic acid	0.35 ± 0.07
*p*-Anisic acid	0.18 ± 0.00
Secoisolariciresinol	0.13 ± 0.02
*p*-Coumaric acid	0.06 ± 0.00 *
Ferulic acid	0.03 ± 0.00 *

Results are expressed as mean ± SD (*n* = 4). Concentration are expressed in μg g^−1^ of dried plant material. * Indicates that the concentration of the identified compound is below the limits of quantification.

**Table 2 molecules-27-01860-t002:** Spectroscopic data of ^1^H-NMR and ^13^C-NMR spectra [500 and 125 MHz, Pyridine-*d_5_*, 50 °C] of the aglycone part of pennogenin tetraglycoside.

No.	^13^C	DEPT-Q135	^1^H (*J* in Hz)
1	38.10	CH_2_	1.78–1.75 [m, 1H]
			1.01–0.96 [m, 1H]
2	30.70	CH_2_	2.11–2.09 [m, 1H]
			1.88–1.81 [m, 1H]
3	70.80	CH	3.89–3.83 [m, 1H]
4	39.95	CH_2_	2.79 [dd, 1H] *J* = 3.0, 13.1
			2.69 [t, 1H] *J* = 12.0
5	141.47	C	--------
6	122.27	CH	5.369–5.360 [m, 1H]
7	32.97	CH_2_	1.98–1.92 [m, 1H]
			1.58–1.52 [m, 1H]
8	32.88	CH	1.67–1.61 [m, 1H]
9	50.85	CH	1.0–0.95 [m, 1H]
10	37.70	C	----------
11	21.50	CH_2_	1.62–1.58 [m, 1H]
			1.55–1.49 [m, 1H]
12	32.61	CH_2_	2.16–2.11 [m, 1H]
			1.55–1.50 [m, 1H]
13	45.60	C	--------
14	53.62	CH	2.08–2.02 [m, 1H]
15	32.29	CH_2_	2.24–2.19[m, 1H]
			1.54–1.48 [m, 1H]
16	90.68	CH	4.44–4.42 [m, 1H]
17	90.71	C	--------
18	17.62	CH_3_	0.96 [s, 3H]
19	19.93	CH_3_	1.08 [s, 3H]
20	45.35	CH	2.28 [q, 1H] *J* = 7.2
21	9.95	CH_3_	1.22 [d, 3H] *J* = 7.2
22	110.39	C	--------
23	32.55	CH_2_	1.77–1.79 [m, 2H]
24	29.31	CH_2_	1.62–1.59 [m, 2H]
25	30.94	CH	1.62–1.59 [m,1H]
26	67.28	CH_2_	3.55–3.49 [m, 2H]
27	17.72	CH_3_	0.71 [d, 3H] *J* = 5.5

Chemical shifts for ^1^H and ^13^C are expressed in ppm using the residual signal of Py-*d*_5_ as reference. The coupling constants (*J*) for proton are expressed in Hertz (Hz). The analyzed spectra were obtained at 50 °C, using water suppression.

**Table 3 molecules-27-01860-t003:** 1H-NMR and 13C-NMR spectroscopic data [500 and 125 MHz, Pyridine d5, 50 °C] of the sugar moieties of the pennogenin tetraglycoside.

**No.**	**^13^C**	**DEPTQ135**	**^1^H (*J* in Hz)**
Glucose			
1	100.94	CH	4.92 [d, 1H] *J* = 7.1
2	78.65	CH	4.14–4.11 [m, 1H]
3	78.24	CH	4.17–4.12 [m, 1H]
4	78.97	CH	4.27–4.23 [m, 1H] *J* = 8.8
5	77.39	CH	3.65–3.62 [m, 1H]
6	61.95	CH_2_	4.20–4.17 [m, 1H]
			4.04 [dd, 1H] *J* = 3.5,12.1
Rhamnose I			
1	102.83	CH	5.69 [s, 1H]
2	73.33	CH	4.51–4.50 [m, 1H]
3	73.63	CH	4.48–4.46 [m, 1H] *J* = 8.65
4	80.87	CH	4.36–4.32 [m, 1H] *J* = 9.25
5	68.98	CH	4.73–4.70 [m, 1H]
6	19.29	CH_3_	1.55 [d, 3H] *J* = 6.2
Rhamnose II			
1	103.62	CH	6.1409 [s, 1H]
2	73.06	CH	4.81–4.80 [m, 1H]
3	73.33	CH	4.44–4.42 [m, 1H] *J* = 3.3, 9.1
4	74.53	CH	4.26–4.22 [m, 1H] *J* = 9.2
5	70.82	CH	4.31–4.26 [m, 1H]
6	18.85	CH_3_	1.58 [d, 3H] *J* = 5.95
Rhamnose III			
1	102.59	CH	6.23 [s, 1H]
2	72.92	CH	4.78–4.76 [m, 1H]
3	73.33	CH	4.57 [dd, 1H] *J* = 3.35, 9.25
4	74.68	CH	4.32–4.28 [m, 1H] *J* = 9.2
5	69.97	CH	4.88–4.85 [m, 1H]
6	19.06	CH_3_	1.74 [d, 3H] *J* = 6.15

Chemical shifts for ^1^H and ^13^C expressed in ppm and with reference to residual signal of Py-d5, as well as the coupling constants (*J*) for proton expressed in Hertz (Hz).

## Data Availability

Experimental data are available on request from the corresponding author.

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
