# Peer review of "Isolation and Identification of Pennogenin Tetraglycoside from Cestrum nocturnum (Solanaceae) and Its Antifungal Activity against Fusarium kuroshium, Causal Agent of Fusarium Dieback"

_molecules, 2022, doi:10.3390/molecules27061860_

Round 1

Reviewer 1 Report

The authors discovered and identified Cestrum nocturnum pennogenin tetraglycoside's antifungal activity against Fusarium kuroshium, the fungus that causes Fusarium dieback. The authors have done a great job with this study. However, before publication, the authors must clarify a few minor revisions.

  1. Please rewrite the result and discussion sections to represent your experiment and results more accurately.
  2. Authors must also improve the quality of all figure legends by giving more details and in all figures, I recommend including a figure legend headline.
  3. For the first time writing, please use abbreviations for all short forms.
  4. Specifically, please update the figure legend in figure 1 to include more information. Include days 3 and 7 information in the legend. What do you mean by C+ and C-? Please add more information so that readers can understand what you're saying.
  5. Please add concentration to the x-axis in figure 2(a) as well. In this experiment, how many replicates (n) were employed?
  6. In figure legend 3, there is no explanation of (h) and two times authors used (e). I recommend that you read over what you've written again.
  7. There should be no full stop (.) in any table headline.

Author Response

  • Reviewer 1

Comments and Suggestions for Authors

The authors discovered and identified Cestrum nocturnum pennogenin tetraglycoside's antifungal activity against Fusarium kuroshium, the fungus that causes Fusarium dieback. The authors have done a great job with this study. However, before publication, the authors must clarify a few minor revisions.

We truly appreciate the encouraging comments of the reviewer. We have considered all her/his detailed suggestions in preparing our revised Ms. We hope that our responses will be deemed satisfactory.

Query 1. Please rewrite the result and discussion sections to represent your experiment and results more accurately.

We appreciate the comment. All Ms sections, with special emphasis in result and discussion, were revised and rewritten to explain the experiments and results more accurately and precise.

Query 2. Authors must also improve the quality of all figure legends by giving more details and in all figures, I recommend including a figure legend headline.

We thank the reviewer for this comment. We apologize for the omission. All figure legends were improved, adding important details and headlines as it was suggested.

Query 3. For the first time writing, please use abbreviations for all short forms.

We appreciate the reviewer comment. We revised throughout of our Ms the correct use of abbreviations.

 Query 4. Specifically, please update the figure legend in figure 1 to include more information. Include days 3 and 7 information in the legend. What do you mean by C+ and C-? Please add more information so that readers can understand what you're saying.

We thank the reviewer for pointing out this observation. Consequently, in the revised Ms we have included an improved version of Figure 1 and added the meaning of C+ and C-.

Query 5. Please add concentration to the x-axis in figure 2(a) as well. In this experiment, how many replicates (n) were employed?

We thank Reviewer 1 for this comment and apologize for the omission of including important information. We followed the reviewer's suggestion and now provide more details in the figure legends. In Figure 2, the concentrations evaluated were included on the x axis as it was suggested.

Query 6. In figure legend 3, there is no explanation of (h) and two times authors used (e). I recommend that you read over what you've written again.

We appreciate this reviewer observation and apologize for the omission. Now it is included an explanation about (h) in the figure legend 3, and we have double checked our revised Ms to avoid again this kind of unvoluntary mistakes.

Query 7. There should be no full stop (.) in any table headline.

We appreciate the comment. There is not any full stop on any table headline.

Reviewer 2 Report

The article concerns the antifungal activity of extracts and the glycoside obtained from them against Fusarium solani and Fusarium kuroshium.

The authors are the first to describe the presence in the plant of Cestrum nocturnum a known compound pennogenin tetraglycoside and show its high ability to inhibit the growth of two strains of Fusaria.

The research on F. kuroshium is particularly interesting, because these fungi in symbosis with the insect Euwallacea kuroshio cause great damage to forestry and agriculture.

In my opinion, the results of the research on Fusarium solani were included mainly with the intention of increasing the amount of data entitling the article to be considered a "full paper". The idea of ​​preceding the tests on F. kuroshium with tests for F. solani (line 104) is not very convincing, because the authors did not select all the subfractions with the highest activity towards F. solani for further studies on F. kuroshium Fig. 2a, e.g. they did not take into account the ECn subfraction -F4-6 or ECn-F4-20. However, in tests assessing the possibility of lowering the concentration of active fractions, without explaining the assumptions of the experiment, they examined the ECn-F4-20 and ECn-F4-27 subfractions. Moreover, the most active compound was found in the fraction with low activity against F. solani.

Please explain the test description (line 116). Was fraction ECn-F4-17 formed from the combination of eluates 181-210 and then split into fractions 25-31? If so, what were the separation conditions?

If not, it is worth testing the ECn-F4-17 fraction, as it contains a compound / compounds with high antifungal activity other than pennogenin tetraglycoside.

The natural culmination of the performed experiments would be the determination of the IC50 for the ECn-F4 fraction and the combined subfractions ECn-F4-25 - ECn-F4-31 (ECn-F4-17 ???) and pure pennogenin tetraglycoside. This would allow to assess whether it is worth purify pennogenin tetraglycoside and whether C. nocturnum preparations have any application potential.

other comments

- lines 359-353: the ECn-5 fraction was not formed as a result of "liquid-liquid extraction", but rather solid state-liquid extraction or maceration, at the same time water cannot be called an organic phase

- the strain number of F. solani was not provided

- there are typos in the list of authors and the list of authors is not identical to "citation".

Author Response

  • Reviewer 2

Comments and Suggestions for Authors

The article concerns the antifungal activity of extracts and the glycoside obtained from them against Fusarium solani and Fusarium kuroshium.

The authors are the first to describe the presence in the plant of Cestrum nocturnum a known compound pennogenin tetraglycoside and show its high ability to inhibit the growth of two strains of Fusaria.

The research on F. kuroshium is particularly interesting, because these fungi in symbosis with the insect Euwallacea kuroshio cause great damage to forestry and agriculture.

We are glad that the referee found our study interesting and appreciate his/her encouraging comments.

Query 1. In my opinion, the results of the research on Fusarium solani were included mainly with the intention of increasing the amount of data entitling the article to be considered a "full paper". The idea of ​​preceding the tests on F. kuroshium with tests for F. solani (line 104) is not very convincing, because the authors did not select all the subfractions with the highest activity towards F. solani for further studies on F. kuroshium Fig. 2a, e.g. they did not take into account the ECn subfraction -F4-6 or ECn-F4-20. However, in tests assessing the possibility of lowering the concentration of active fractions, without explaining the assumptions of the experiment, they examined the ECn-F4-20 and ECn-F4-27 subfractions. Moreover, the most active compound was found in the fraction with low activity against F. solani.

We thank Reviewer 2 for pointing out these observations of our work. Nonetheless, we respectfully disagree with the reviewer on her/his statement about that F. solani assays is to increase the amount of data of our manuscript.  The evaluation of the antifungal activity against F. solani has been very important to us as a first approach to address F. kuroshium. As it has been explained in our Ms and reported previously by part of our research group (Guevara-Avendaño et al., 2019; Báez-Vallejo et al., 2020) in biotesting bacterial natural products as potential biocontrol agents, we are not able to perform experiments in our laboratory with F. kuroshium on regular basis, since this phytopathogen is a quarantined pest and requires special permits and biosecurity facilities for its manipulation. The National Service for Agrifood Health, Safety and Quality of the Mexican Government allowed us to perform limited and very punctual experiments in its diagnostic laboratories located in a different state where we are in INECOL. So, to make our research more efficient under these circumstances, and increasing the probabilities of finding active subfractions, we needed previous information and we performed previous analyses with F. solani as microbiological model due its close phylogenetic relationship to F. kuroshium. However, it is true that we evaluated active and non-active secondary fractions with F. kuroshium and the reason was that we selected the fractions considering also the available amount of fractions obtained from the second fractionation along with the bioactivity results from the antifungal assays against F. solani. This point was declared in the revised version of our Ms. In addition, we grouped the secondary fractions (ECn-F4-25 to ECn-F4-31) which correspond to the same precipitate obtained in different but sequential eluates and corrected some mistakes in the names of some secondary fractions. We sincerely believe that the revised version is now clearer and more precise. We apologize for the confusion that this might cause during the revision of our original Ms.

References:

  1. Guevara-Avendaño, E.; Bejarano-Bolívar, A.A.; Kiel-Martínez, A.L.; Ramírez-Vázquez, M.; Méndez-Bravo, A.; von Wobeser, E.A.; Sánchez-Rangel, D.; Guerrero-Analco, J.A.; Eskalen, A.; Reverchon, F. Avocado rhizobacteria emit volatile organic compounds with antifungal activity against Fusarium solani, Fusarium associated with Kuroshio shot hole borer, and Colletotrichum gloeosporioides. Microbiol. Res. 2019, 219, 74–83, doi.org/10.1016/j.micres.2018.11.009.
  2. Báez-Vallejo, N.; Camarena-Pozos, D.A.; Monribot-Villanueva, J.L.; Ramírez-Vázquez, M.; Carrión-Villarnovo, G.L.; Guerrero-Analco, J.A.;Partida-Martínez, L.P.; Reverchon, F. Forest tree associated bacteria for potential biological control of Fusarium solani and of Fusarium kuroshium, causal agent of Fusarium dieback. Microbiol. Res. 2020,235, 126440, doi:10.1016/j.micres.2020.126440

Query 2. Please explain the test description (line 116). Was fraction ECn-F4-17 formed from the combination of eluates 181-210 and then split into fractions 25-31? If so, what were the separation conditions?

If not, it is worth testing the ECn-F4-17 fraction, as it contains a compound / compounds with high antifungal activity other than pennogenin tetraglycoside.

We thank Reviewer 2 for pointing out this observation. As it was described in our Ms, an amorphous precipitate was obtained from eluates 181-210 and grouped in seven secondary fractions (25-31) based on thin layer chromatographic analyses. However, the major component present in these secondary fractions was the same pennogenin tetraglycoside compound. So, in the revised version and aiming to clarify this point, we decided to show only as secondary fraction of this group to ECN-F4-25 that was further analyzed by spectroscopic and spectrometric approaches and identify as pennogenin tetraglycoside. As mentioned above, we apologize for the confusion that this caused during the revision of our original Ms. 

Query 3. The natural culmination of the performed experiments would be the determination of the IC50 for the ECn-F4 fraction and the combined subfractions ECn-F4-25 - ECn-F4-31 (ECn-F4-17 ???) and pure pennogenin tetraglycoside. This would allow to assess whether it is worth purify pennogenin tetraglycoside and whether C. nocturnum preparations have any application potential.

We highly appreciate the reviewer comment which encourages us to perform additional experiments and go deeper in the antifungal activity of the pennogenin tetraglycoside. The goal of the present Ms is to show for the very first time the potential of this compound as an antifungal plant agent against an emergent phytopathogen. As we mentioned previously, we need access to the biosecurity facilities from the National Service for Agrifood Health, Safety and Quality from Mexican government to carry out more experiments. Unfortunately, the access to these facilities for research purposes has been shut down since March 2020 as consequence of the COVID-19 pandemic. Nonetheless, we truly believe that the current results of our Ms contribute with novel knowledge on the chemical and biological potential of C. nocturnum.

Other comments

Query 4. - lines 359-353: the ECn-5 fraction was not formed as a result of "liquid-liquid extraction", but rather solid state-liquid extraction or maceration, at the same time water cannot be called an organic phase.

We thank Reviewer 2 for this comment. Respectfully, we still considering that the chemical composition of ECn-F5 is the result of liquid-liquid extractions performed with all the organic solvents (hexane, dichloromethane, ethyl acetate and butanol). However, we agreed with the reviewer that ECn-F5 cannot be called an organic phase. In fact, we declare in the same lines that the aqueous phase was lyophilized to dryness. We changed the paragraph in the revised version of our Ms to clarify this point.

Query 5. - the strain number of F. solani was not provided

We apologize for the omission. We added the strain number for the collection of the High Technology Laboratory of Xalapa which belongs to the University of Veracruz, Mexico.

Query 6. - there are typos in the list of authors and the list of authors is not identical to "citation".

We apologize for the mistakes. We revised all Ms sections and corrected the typos we found, including the “citation”.

Reviewer 3 Report

The work presented is very interesting and the characterization of pennogenin tetraglycoside is clearly described. As said by the authors, in literature the characterization of this saponin is lacking and even more its antifungal activity.

Some points to underline:

-in line 109 the extracts ECn-F4-20 and ECn-F4-27 are mentioned, but they are not present in the Figure 2a. Please check it.

-in Figure 2b, the extract ECn-F4-20, not shown in Figure 2a, was tested at different concentrations on fusarium kuroshium . Please clarify this point.

Some typos:

-in line 190 the quaternary carbon at 141.47 ppm should be attributed to C5 of the molecule and not C6 as described in the paper.
-in line 202 the word "asssigned" should be corrected as well as in line 216 "monosacharides", in line 296 the word "well", in line 285 the word "mecanism"; in line 384 the word "micelyal".

Overall the work is well done and the one- and two-dimensional NMR studies are very accurate.

Author Response

  • Reviewer 3

Comments and Suggestions for Authors

The work presented is very interesting and the characterization of pennogenin tetraglycoside is clearly described. As said by the authors, in literature the characterization of this saponin is lacking and even more its antifungal activity.

We truly appreciate the encouraging comments of the reviewer to our manuscript.

Some points to underline:

Query 1. -in line 109 the extracts ECn-F4-20 and ECn-F4-27 are mentioned, but they are not present in the Figure 2a. Please check it.

We thank Reviewer 3 for pointing out this observation. We apologize for the unintentional mistakes. In the revised version of our Ms we corrected mistakes in the names of some secondary fractions. Regarding ECn-F4-27, we decided to group the secondary fractions (ECn-F4-25 to ECn-F4-31) which correspond to the same precipitate obtained in different but sequential eluates. We decided only to show the secondary fraction (ECN-F4-25) that was further analyzed by spectroscopic and spectrometric approaches and identified as the active pennogenin tetraglycoside compound.

Query 2. -in Figure 2b, the extract ECn-F4-20, not shown in Figure 2a, was tested at different concentrations on Fusarium kuroshium . Please clarify this point.

We thank Reviewer for this comment. As it was suggested, the names of the secondary fractions were revised and corrected to clarify the point.

Some typos:

 Query 3. -in line 190 the quaternary carbon at 141.47 ppm should be attributed to C5 of the molecule and not C6 as described in the paper.

We thank Reviewer 3 for this comment. In concordance, we corrected this mistake in the revised version of our Ms.

Query 4. -in line 202 the word "asssigned" should be corrected as well as in line 216 "monosacharides", in line 296 the word "well", in line 285 the word "mecanism"; in line 384 the word "micelyal".

We thank the Reviewer for pointing out these corrections. We addressed all in the revised version of our Ms as it was suggested.

Overall the work is well done and the one- and two-dimensional NMR studies are very accurate.

Again, we thank the Reviewer for her/his positive criticism to our work.

Reviewer 4 Report

This is an interesting work that described for the first time (as the authors claim) the effect of the crude extract, fractions and one pure compound from Cestrum nocturnum on Fusarium kuroshium and F. solani mycelial growth inhibition. There were some comments in the following:

line 31: leaves / leaves, please change it.

line 45: Persea americana Mill / Persea americana, please change it.

line 57: Cestrum nocturnum L / Cestrum nocturnum, please change it.

line 46-56: Could you please explain the relationship for “fungus-beetle complex”? You only mentioned “there is no effective method to control the spread of this complex pest”, the issue is not related to Fusarium kuroshium. If we try to control F. kuroshium, would it have great effect on “fungus-beetle complex”?

line 104: “Based on the results from the antifungal activity assay against F. solani”---could you please explain more clearly?

line 137: Why did you choose ECn-F4-16,18 and 20 for SEM observation? The author should show the concentration of ECn at the same time.

line 375: how to filter the mycelium?

line 388: how to measure the diameter of fungal mycelia, with some software? You should show us the mycelial growth inhibition photos.

Author Response

  • Reviewer 4

Comments and Suggestions for Authors

This is an interesting work that described for the first time (as the authors claim) the effect of the crude extract, fractions and one pure compound from Cestrum nocturnum on Fusarium kuroshium and F. solani mycelial growth inhibition. There were some comments in the following:
We thank the reviewer for highlighting the important aspects of our study.

 Query 1. line 31: leaves / leaves, please change it.

We thank Reviewer 4 for pointing out this correction.

Query 2. line 45: Persea americana Mill / Persea americana, please change it.

We thank Reviewer 4 for pointing out this correction.

Query 3. line 57: Cestrum nocturnum L / Cestrum nocturnum, please change it.

We thank Reviewer 4 for pointing out  this correction.

Query 4. line 46-56: Could you please explain the relationship for “fungus-beetle complex”? You only mentioned “there is no effective method to control the spread of this complex pest”, the issue is not related to Fusarium kuroshium. If we try to control F. kuroshium, would it have great effect on “fungus-beetle complex”?

We thank Reviewer 4 for this comment. We believe that reviewer’s suggestion is certainly appropriate, and we thank her/him for the opportunity and guidance to clarify our message. Particularly, in the Introduction section of the revised version of our Ms, we have addressed the “fungus-beetle complex” in more detail. Different approaches have been used trying to control this pest complex. Some research groups are trying to control the beetle and others (including us) are trying to control the fungus. We believe that discovering antifungal molecules will help in the coordinated effort against this fungus-beetle complex.

Query 5. line 104: “Based on the results from the antifungal activity assay against F. solani”---could you please explain more clearly?

We thank the reviewer for highlighting this point. In the revised Ms we changed this sentence to explain more clearly. Now it can be read in the revised version: “Considering the obtained amount for each subfraction from the second fractionation as well as the results from the antifungal activity assay against F. solani (Figure 1), seven secondary fractions were selected for further evaluation against F. kuroshium in the Mexican National Phytosanitary Reference Center (CNRF by its acronym in Spanish).”

Query 6. line 137: Why did you choose ECn-F4-16,18 and 20 for SEM observation? The author should show the concentration of ECn at the same time. The secondary fractions selected for SEM observation were ECn-F4-16 at 0.5 mg mL-1, ECn-F4-18 at 0.2 mg mL-1 and ECn-F4-20 at 0.5 mg mL-1. The selection was based on the results exhibited by these secondary fractions and concentrations on F. kuroshium mycelial growth inhibition (Figure 2). We choose secondary fractions and concentrations that caused a moderate damage. ECn-F4-16 at 0.5 mg mL-1 caused around 60% of growth inhibition, ECn-F4-18 at 0.2 mg mL-1 caused around 80% of growth inhibition and ECn-F4-20 at 0.5 mg mL-1 caused around 70% of growth inhibition. The methanolic extract (ECn) before any fractionation was not observed by SEM, neither the precipitate (ECn-F4-25, the pennogenin tetraglycoside). We appreciate the comment of reviewer 4 because it encourages us to perform further experiments and go deeper in the antifungal activity of the pennogenin tetraglycoside. Nonetheless, the limiting aspect is that we are not able to perform experiments in our laboratory with F. kuroshium, because it is a quarantined fungal phytophatogen and requires special permit and biosecurity facilities for its manipulation and these are provided by The National Service for Agrifood Health, Safety and Quality of Mexican government. Unfortunately, the access to these facilities for research purposes has been shut down since March 2020 as consequence of the COVID-19 pandemic. Nonetheless, we truly believe that the current results of our Ms contribute with novel knowledge on the chemical and biological potential of C. nocturnum.  We made changes in the revised Ms to clarify this point

Query 7. line 375: how to filter the mycelium?

We thank Reviewer 4 for this very important observation. In fact, there was an error in the writing of the methodology, the spore solution was not filtered. The statement was removed from the materials and methods section.

Query 8. line 388: how to measure the diameter of fungal mycelia, with some software? You should show us the mycelial growth inhibition photos.

We thank Reviewer 4 for this observation. Mycelial growth diameter was measured using Mycometre software 2.01 which allow us to make measurements of the fungal mycelia diameter previous calibration. In addition, representative photos of the antifungal tests were incorporated in the supplementary material as it was suggested.

Round 2

Reviewer 2 Report

The Authors improved the manuscript in many respects, however the presented research results require the determination of the MIC50. This is the only way to quantify the activity of a chemical compound or plant extract. Only then will other researchers be able to compare the activity of the new agents against F. kuroshium with the achievements of the Authors of this article.
I understand the troubles of the COVID-19 pandemic that we all struggle with, but it is worth waiting for the possibility of additional research to make the article fully valuable.

Author Response

Comments and Suggestions for Authors

The Authors improved the manuscript in many respects, however the presented research results require the determination of the MIC50. This is the only way to quantify the activity of a chemical compound or plant extract. Only then will other researchers be able to compare the activity of the new agents against F. kuroshium with the achievements of the Authors of this article.
I understand the troubles of the COVID-19 pandemic that we all struggle with, but it is worth waiting for the possibility of additional research to make the article fully valuable.

We thank the reviewer for her/his comment. We have always considered the determination of the MIC50 to be of great importance. Actually, we tested secondary fractions at different concentrations (0.2, 0.5, 1 and 2 mg ml-1) in order to determine MIC50. Unfortunately, from a technical point of view, but fortunately biologically, the mycelial growth inhibition exhibited by pennogenin tetraglycoside was so high that it did not allow us to calculate MIC50. As we mentioned above, we need access to the biosafety facilities of the Mexican government’s National Service for Health, Safety and Food Quality to conduct additional experiments. Unfortunately, access to these facilities for research purposes has been shut down since March 2020 as a result of the COVID-19 pandemic. However, we sincerely believe that the results of our Ms in the form that is currently presented provide new insights into the chemical and biological potential of C. nocturnum and its compounds against an emergent phytopathogen. We consider that the determination of the MIC50 would be additional data that would not crucially modify the significance of obtained results. Additionally, we believe that the results of antifungal activity expressed in % of mycelial growth inhibition are comparable, since in the literature, even in specialized microbiology journals, there is a large number of recent articles that express their results in the same way as we are reporting ours.

Respectfully, in what follows we have listed at least 26 articles published only in Molecules from 2020 (Table 1) to date, plus three articles published in different journals by our working group and colleagues (Table 2), where MIC determination was not determinant for these contributions to be considered worth of publication. We deeply thank you for criticism that it has always been taken as positive. Please see attached document.

Reviewer 4 Report

It is much better. I think the authors bring the manuscript to a publication level.

Author Response

  • Reviewer 4

Comments and Suggestions for Authors

It is much better. I think the authors bring the manuscript to a publication level.

We truly appreciate the encouraging comments of the reviewer 4 to our manuscript.
